# Intrauterine Zika virus infection of pregnant immunocompetent mice models transplacental transmission and adverse perinatal outcomes

Meghan S. Vermillion[1,2], Jun Lei[3], Yahya Shabi[3], Victoria K. Baxter[1,2], Nathan P. Crilly[1,†], Michael McLane[3], Diane E. Griffin[1], Andrew Pekosz[1], Sabra L. Klein[1] & Irina Burd[3]

Zika virus (ZIKV) crosses the placenta and causes congenital disease. Here we develop an animal model utilizing direct ZIKV inoculation into the uterine wall of pregnant, immuno-competent mice to evaluate transplacental transmission. Intrauterine inoculation at embryonic day (E) 10, but not E14, with African, Asian or American strains of ZIKV reduces fetal viability and increases infection of placental and fetal tissues. ZIKV inoculation at E10 causes placental inflammation, placental dysfunction and reduces neonatal brain cortical thickness, which is associated with increased activation of microglia. Viral antigen localizes in trophoblast and endothelial cells in the placenta, and endothelial, microglial and neural progenitor cells in the fetal brain. ZIKV infection of the placenta increases production of IFNβ and expression of IFN-stimulated genes 48 h after infection. This mouse model provides a platform for identifying factors at the maternal–fetal interface that contribute to adverse perinatal outcomes in a host with an intact immune system.

[1] W. Harry Feinstone Department of Molecular Microbiology and Immunology, Johns Hopkins Bloomberg School of Public Health, Baltimore, Maryland 21205, USA. [2] Department of Molecular and Comparative Pathobiology, The Johns Hopkins School of Medicine, Baltimore, Maryland 21205, USA. [3] Integrated Research Center for Fetal Medicine, Department of Gynecology and Obstetrics, Johns Hopkins University School of Medicine, Baltimore, Maryland 21287, USA. † Present address: University of Tennessee School of Veterinary Medicine, Knoxville, Tennessee 37996, USA. Correspondence and requests for materials should be addressed to S.L.K. (email: Sklein2@jhu.edu) or to I.B. (email: Iburd@jhmi.edu).

Zika virus (ZIKV) infection of pregnant women is linked with a variety of birth defects including microcephaly, intracranial calcifications, ocular disease, hearing deficits, intrauterine growth restriction and spontaneous abortion, collectively termed congenital Zika syndrome[1–7], and ZIKV RNA has been detected in amniotic fluid from affected fetuses, and in the brains of fetuses from terminated pregnancies[2–5,8–10]. A recent case-control study has concluded the microcephaly epidemic in Brazil is caused by congenital ZIKV infection[11], but the pathogenesis and virus strain-dependence of ZIKV infection at the maternal–fetal interface has not been well characterized.

The risk to the fetus following maternal ZIKV infection during pregnancy is dependent on gestational age. Infections estimated to have occurred at the end of the first trimester are correlated with higher rates of congenital infection and more severe fetal outcomes than infections occurring later during the second and third trimesters[12–14]. Vaginal infection with ZIKV in immuno-compromised mice has also demonstrated gestational age-dependent differences in effects on fetal body mass[15]. However, no animal model to date has allowed investigation of how gestational age, and more specifically how the timing of placentation (that is, formation of the cellular barrier that separates maternal and fetal circulations) and fetal neurogenesis influence vertical ZIKV transmission.

In part because ZIKV NS5 effectively targets the interferon (IFN) signalling pathway in humans, but not in mice[16], mice are less susceptible to infection, and immunocompetent adult mice replicate only low levels of viral RNA and develop no evidence of clinical disease[15,17]. Published murine models of ZIKV infection and fetal transmission have predominately studied pathogenesis following systemic routes of virus inoculation (for example, subcutaneous, intraperitoneal or intravenous), comparing immu-nocompetent with immunocompromised IFN receptor deficient (IFNAR$^{-/-}$) mice[15,18]. ZIKV infection of IFNAR$^{-/-}$ mice results in productive virus replication and clinical disease[17,19,20]. ZIKV infection of IFNAR$^{-/-}$ and other immunodeficient mice during early pregnancy, that is before complete placentation[21], results in virus transmission to the fetus[15,22]. Because IFN signalling has been identified as an important factor influencing ZIKV infection in vitro[23–26], an accurate in vivo ZIKV model of maternal–fetal transmission should take into account the IFN responses induced in the maternal and fetal tissues after infection. ZIKV infection of pregnant, immunocompetent mice has been limited to SJL and C57BL/6 mice, with the outcomes following systemic or vaginal routes of infection being variable, with copies of viral RNA being very low, and in some cases requiring the use of viral inoculums that are several logs higher than the amount of virus subsequently detected in either reproductive tissues or fetuses[14,15,22,27]. Previous studies also do not adequately address ZIKV transmission across the placental barrier as most studies have relied on infection of dams before complete place-ntation[15,18].

There is an ongoing and urgent need to develop and characterize animal models for the study of ZIKV maternal–fetal transmission and disease, and to identify factors at the maternal–fetal interface that contribute to adverse perinatal outcomes. We sought to develop an animal model using immunocompetent, outbred, pregnant mice that would enable us to evaluate the mechanisms of ZIKV transmission at the maternal–fetal interface (that is, across the placental barrier). Using direct intrauterine (IU) inoculation at times during gestation, when the placental barrier is formed[21], we bypassed the need for peripheral ZIKV replication in murine tissues and determined the effects of the route of inoculation, gestational age, and ZIKV strain on perinatal outcomes. This tractable system provides a model for the study of vertical ZIKV transmission and

investigation of the host and viral determinants that influence pathogenesis at the maternal–fetal interface in an immuno-competent host.

## Results

**Placental ZIKV infection has perinatal effects.** To test whether IU ZIKV inoculation results in ZIKV infection during pregnancy, immunocompetent, timed-pregnant mice were inoculated at E10 with either $10^6$ TCID$_{50}$ units of 1968 Nigerian ZIKV (IBH 30656) or vehicle by direct IU injections or systemic intraperitoneal (IP) injections (Fig. 1a). Because placental development in the mouse is delayed compared with humans and the definitive placental structure is not present until E10 in mice[21], which is comparable to gestational day 21 in humans, we chose E10 as our earliest infection time-point. Tissues from dams and fetuses were collected 48 h post-inoculation (hpi). Compared with vehicle-infected controls, fetal viability at 48 hpi was significantly reduced following IU, but not IP, ZIKV inoculation (Fig. 1b). ZIKV RNA—indicating the presence of either infectious or noninfectious virus—was not detectable in serum, but was detected at low-levels in the spleens of both IU and IP inoculated dams (Fig. 1c). Intrauterine, but not IP, inoculation with ZIKV resulted in low levels of viral RNA in the uterine horns, and higher levels of viral RNA in placentas and fetal heads (Fig. 1c). Further, there was a significant association between levels of viral RNA in placentas and corresponding fetal heads (Fig. 1d). Infectious virus—indicating an active, ongoing infection—was detected in uterine horns, placental and fetal heads of IU, but not IP, inoculated dams (Fig. 1e). In fetal heads collected from IU-infected dams, there was a significant association between the amount of viral RNA and infectious virus (Fig. 1f).

To determine if contemporary strains of ZIKV could also be transmitted at the maternal–fetal interface, timed-pregnant mice were inoculated IU at E10 with either vehicle or $10^6$ TCID$_{50}$ units of 2010 Cambodian ZIKV (FSS13025), 2015 Brazilian ZIKV (Pariaba) or 2015 Puerto Rican ZIKV (PRVABC59), as well as 1968 Nigerian ZIKV, and tissues from dams and fetuses were collected 48 hpi. Fetal viability at 48 h following infection with either historic (1968 Nigerian) or contemporary (2010 Cambo-dian, 2015 Brazilian or 2015 Puerto Rican) strains of ZIKV was significantly reduced compared with vehicle-inoculated dams, with no significant difference in fetal viability between ZIKV strains (Fig. 2a). ZIKV RNA was detectable at similar levels in the maternal spleen, uterine horns, placenta and fetal heads, regardless of ZIKV strain (Fig. 2b). Furthermore, the relative viral RNA load (for all ZIKV strains) in the placenta, but not in the maternal spleen, correlated with the corresponding viral load in the fetal heads (Fig. 2c,d). These data illustrate that following IU inoculation of immunocompetent mice, multiple strains of ZIKV have the capacity to induce fetal resorption and productively replicate in both placental and fetal brain tissue, with the presence of ZIKV virus in the placenta predicting transmission to the fetus in this model.

**Fetal outcomes are dependent on gestational age.** Human reports and mouse models suggest that the relative risk to the fetus following ZIKV infection during pregnancy is dependent on gestational age[12,13,15]. We investigated how gestational age at the time of ZIKV exposure affects rates of infection and transmission at the maternal–fetal interface in mice. We compared infection at E10 (that is, the time of complete placentation[21]) with infection at E14, which marks the cessation of placental expansion and growth in mice[28], and corresponds with the third trimester in humans. Following IU inoculation with ZIKV (using the 1968

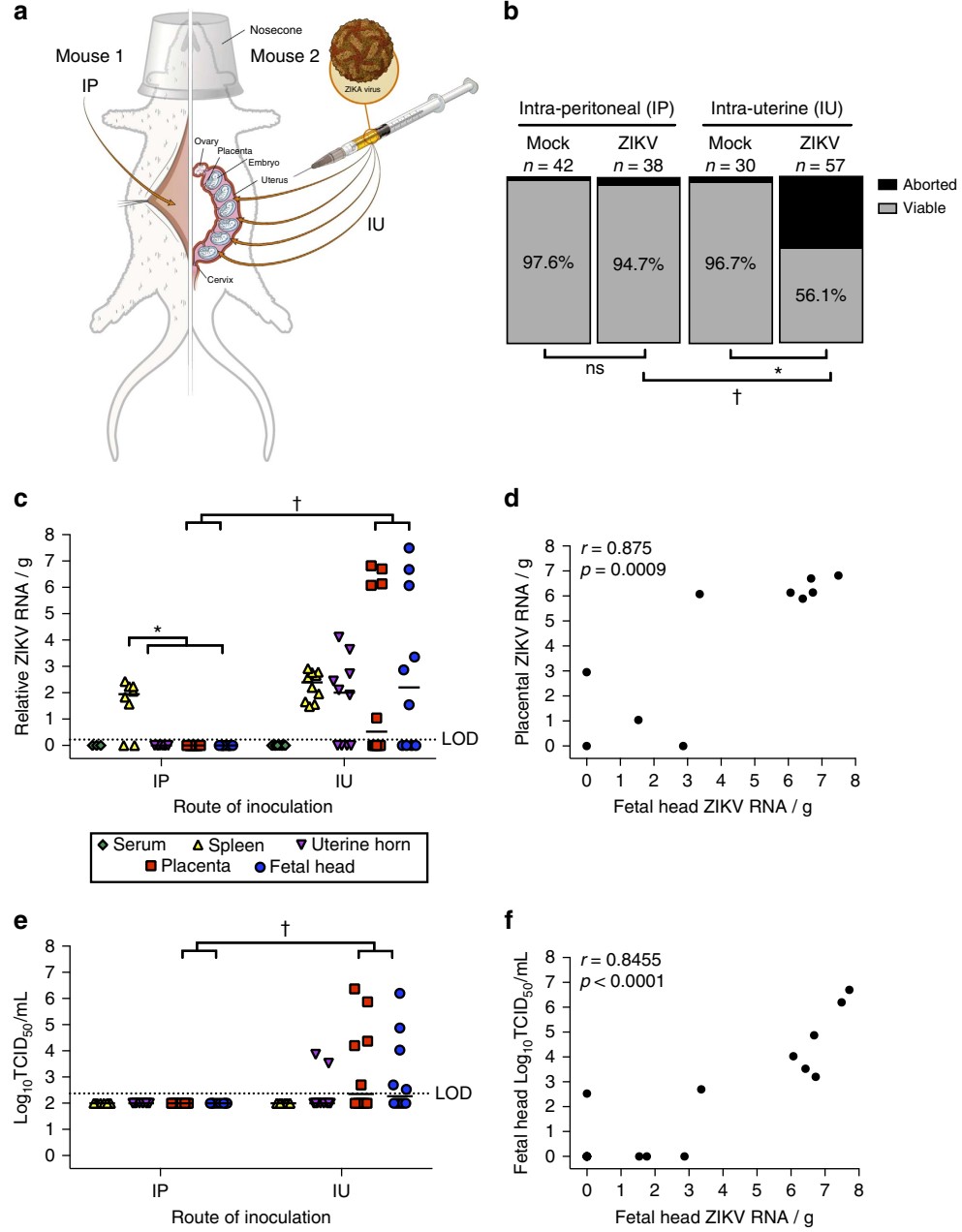

**Figure 1 | Intrauterine (IU) inoculation of ZIKV during pregnancy results in productive infection and altered fetal outcomes.** (**a**) At embryonic day (E) 10, pregnant CD-1 mice underwent a mini-laparotomy in the lower abdomen for inoculation of either $10^6$ TCID$_{50}$ units of ZIKV (1968 Nigeria) or mock infection (IP, $n = 3$ dams; IU, $n = 4$ dams) by intraperitoneal (IP, $n = 8$ dams) or intrauterine (IU, $n = 10$ dams) routes. (**b**) At 48 h post-inoculation (hpi; E12), dams were killed, and fetal viability was determined as the percentage of fetuses within the inoculated uterine horn for ZIKV- and mock-infected dams that were viable. (**c**) ZIKV RNA, determined by qRT-PCR, was quantified from maternal serum, spleen, uterine horns, placenta and fetal brain tissues collected 48 hpi, with viral RNA in the placenta correlating with viral RNA in fetal tissues collected from IU ZIKV-inoculated dams (**d**). Infectious virus, determined by 50% tissue culture infectious dose (TCID50) (**e**), was quantified from maternal spleen, uterine horns, placenta and fetal brain tissues collected 48 hpi and correlated with viral RNA in fetal heads (**f**). For **c,e**, the median is indicated by the solid line for each tissue and the limits of detection (LOD) are indicated with dashed lines. Chi square test (**b**), two-way ANOVA with Bonferroni *post-hoc* correction (**c,e**), and Spearman correlation analysis (**d,f**), * = significant difference at $P < 0.05$ within a route of inoculation, † = significant difference at $P < 0.05$ between routes of inoculation.

Nigerian strain as a proxy for all ZIKV strains) at either E10 or E14, we compared fetal viability and viral burden at 48 hpi. Fetal viability at 48 hpi was significantly greater in dams infected at E14 compared with those infected at E10 (Fig. 3a). ZIKV RNA was detectable in placentas from E14-inoculated dams, but relative viral RNA loads were lower compared with placentas from

E10-inoculated dams (Fig. 3b), and no infectious virus was detected in placentas from E14-inoculated dams (Fig. 3c). Neither ZIKV RNA nor infectious virus was detectable in fetal heads from E14-inoculated dams, which is in contrast with the readily detectable ZIKV RNA and infectious virus in fetal heads from E10-inoculated dams (Fig. 3b,c). At 48 hpi, although ZIKV RNA

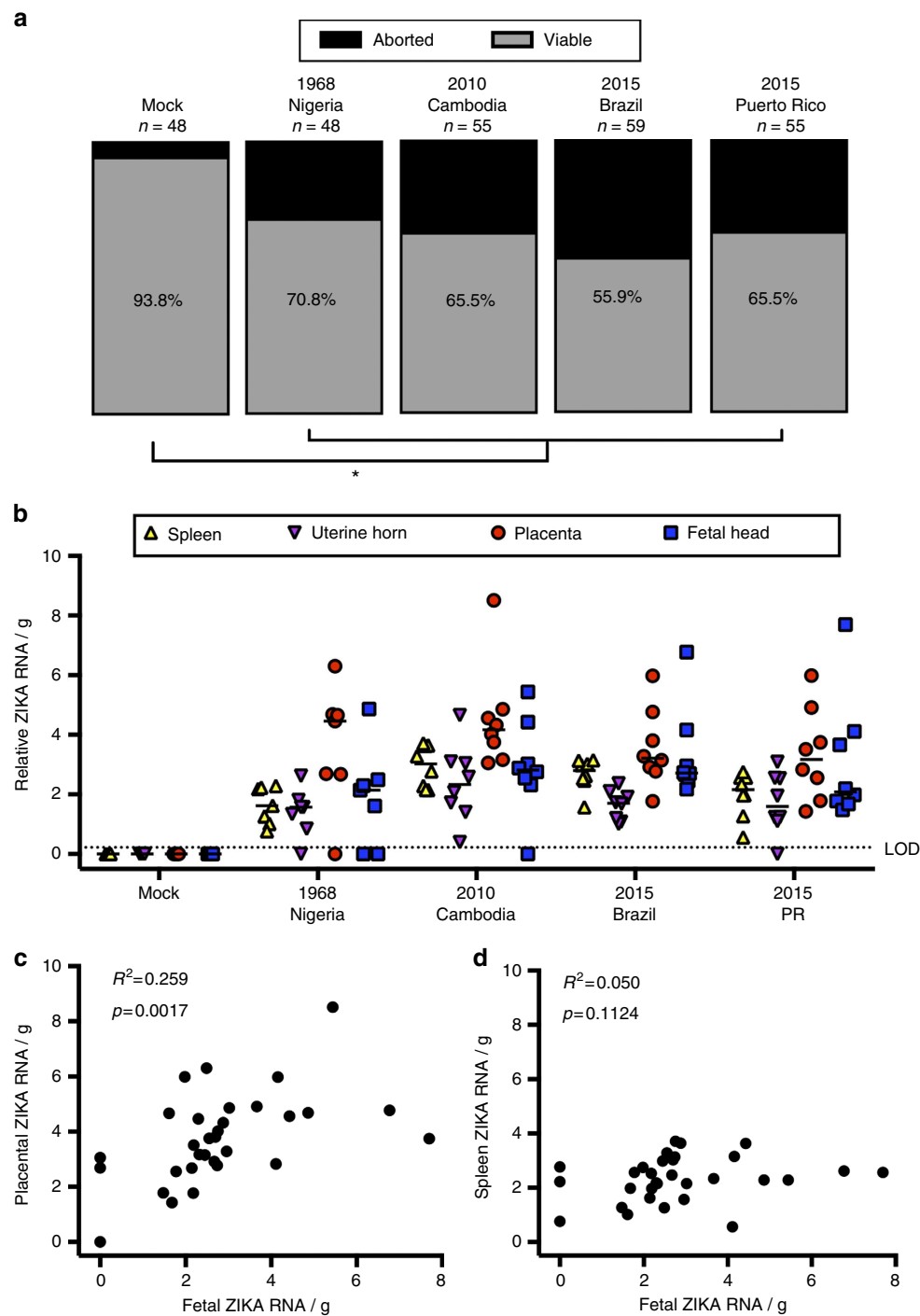

**Figure 2 | Historic and contemporary strains of ZIKV productively infect placental and fetal tissue to alter the outcome of pregnancy.** At embryonic day (E) 10, pregnant CD-1 mice underwent a mini-laparotomy in the lower abdomen for intrauterine (IU) inoculation of $10^6$ TCID$_{50}$ units of the 1968 Nigeria, 2010 Cambodia, 2015 Brazil or 2015 Puerto Rico ZIKV ($n = 7$–8 dams/virus strain) or vehicle ($n = 7$ dams). (**a**) At 48 h post-inoculation (hpi; E12), dams were killed and fetal viability was determined as the percentage of fetuses within the inoculated uterine horn for ZIKV- and mock-infected dams that were viable. (**b**) ZIKV RNA was quantified from maternal spleen, uterine horns, placenta and fetal brain tissues collected 48 hpi, with the median indicated by the solid line for each tissue and the limit of detection (LOD) indicated with a dashed line. (**c**) Viral RNA in the fetus correlated with viral RNA in the placenta (**c**), but not in the spleen (**d**). Chi square test (**a**), two-way ANOVA with Bonferroni *post-hoc* correction (**b**), and Pearson correlation analysis (**c,d**), * = significant difference at $P < 0.05$ between ZIKV- and mock-infection.

was present in maternal spleens and placentas from E14-inoculated dams, infectious virus was not detected in these tissues (Fig. 3b,c).

To evaluate virus replication kinetics and the effect of additional time on fetal viability, we inoculated dams with ZIKV at either E10 or E14, this time killing dams at 96 hpi. Fetal viability at 96 hpi following E10 or E14 inoculations with ZIKV was similar to that observed at 48 hpi (Fig. 3d), as were patterns of viral RNA (Fig. 3e) and infectious virus (Fig. 3f) detection in the maternal spleen, uterine horn, placenta and fetal head, in which

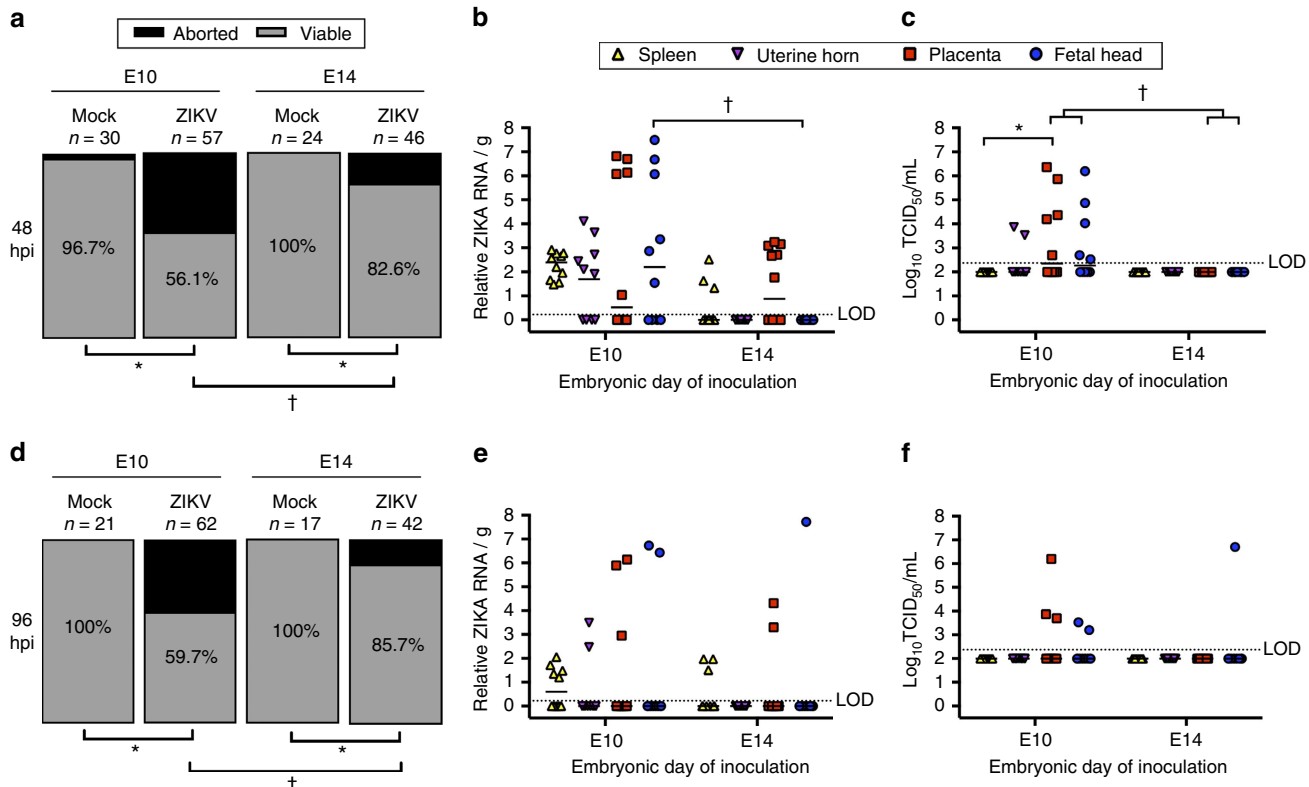

**Figure 3 | The outcome of ZIKV infection during pregnancy depends on gestational age.** At either embryonic day (E) 10 or E14, pregnant CD-1 mice underwent a mini-laparotomy in the lower abdomen for intrauterine (IU) inoculation of ZIKV (1968 Nigeria) or vehicle. (**a**) At 48 h post-inoculation (dpi), dams were killed and fetal viability was determined as the percentage of fetuses within the inoculated uterine horn for ZIKV- and mock-infected dams that were viable ($n = 6$ mock-infected; $n = 10$ E10-inoculated; $n = 12$ E14-inoculated). (**b**) ZIKV RNA copies and (**c**) infectious virus were quantified from maternal spleen, uterine horns, placenta and fetal tissues collected 48 hpi from dams inoculated at either E10 or E14. (**d**) At 96 hpi, dams were killed and fetal viability was measured as the percentage of viable fetuses that were present within the uterus ($n = 4$ mock-infected; $n = 10$ E10-inoculated; $n = 9$ E14-inoculated). (**e**) ZIKV RNA copies and (**f**) infectious virus were quantified from maternal spleen, uterine horns, placenta, and fetal tissues collected 48 or 96 hpi from dams inoculated at either E10 or E14. For **b,c,e,f**, the median is indicated by the solid line for each tissue and the limits of detection (LOD) are indicated with dashed lines. Chi square test (**a,d**) and two-way ANOVA with Bonferroni *post-hoc* correction (**b,c,e,f**), $*$ = significant difference at $P < 0.05$ between mock- and ZIKV-infection (**a,d**) and between tissues following ZIKV infection (**b,c,e,f**), $†$ = significant difference at $P < 0.05$ between gestational ages at the time of infection.

ZIKV loads were generally higher in tissues collected from E10- than E14-inoculated dams, but this did not reach statistical significance. Overall, however, the proportion of ZIKV RNA and infectious virus positive tissues was reduced at 96 hpi compared with 48 hpi (Fig. 3e,f) suggesting control of virus replication in these immunocompetent mice (Fig. 3c,f). Taken together, these data demonstrate that ZIKV inoculation earlier in gestation results in greater transmission of virus to the fetus and reduced fetal viability.

**ZIKV antigen localized in the placenta and fetal brain**. The distribution and cell tropism of ZIKV antigen following IU inoculation at E10 was evaluated in the feto-placental unit 'en mass' by immunohistochemistry (IHC). We selected E10 for evaluation of ZIKV cell tropism because this time-point is associated with placentation[21], neurogenesis[29] and high levels of ZIKV infection (Fig. 3). To assess cell tropism at the inoculation site, IHC for ZIKV antigen in α-actin + smooth muscle cells in the uterine myometrium was evaluated. Despite detectable ZIKV RNA (Fig. 1c), viral antigen was not detectable within the uterine myometrium 48 h following IU inoculation (Supplementary Fig. 1a), which is consistent with the low amounts of infectious virus present in these tissues. In the placenta, however, ZIKV

antigen was abundant at both 48 and 96 hpi (Fig. 4a; Supplementary Fig. 1b) and localized predominantly in Cytokeratin + trophoblasts and Vimentin + endothelial cells (Fig. 4b,c). Placentas from dams inoculated with ZIKV at E10 also sustained significant damage, characterized by an overall loss of definition between placental layers, a reduction of the labyrinth, and significant haemorrhage and mixing of maternal and fetal blood (Fig. 4d), suggestive of a compromised trophoblast–endothelial cell barrier.

We only evaluated the cell tropism of ZIKV in fetal heads because fetal bodies harboured lower levels of viral RNA compared with corresponding fetal heads at both 48 and 96 hpi (Fig. 5a). In the fetal brain, ZIKV antigen was present at both 48 and 96 hpi (Fig. 5b and Supplementary Fig. 1c), where it localized in CD34 + fetal endothelial cells, Iba-1 + microglia, and Nestin + neural progenitor cells (Fig. 5c–e). Similar to reports of direct inoculation into the fetal brain *in vivo*[14,30] or infection of cells *in vitro*[22,23,25,27,31–33], these data suggest that ZIKV can infect multiple cell types in both the placenta and fetal brain.

**Type I IFN signalling in placentas**. The prevailing mouse model for evaluating ZIKV pathogenesis involves the use of type I IFN signalling deficient mice, including but not limited to IFNAR$^{-/-}$

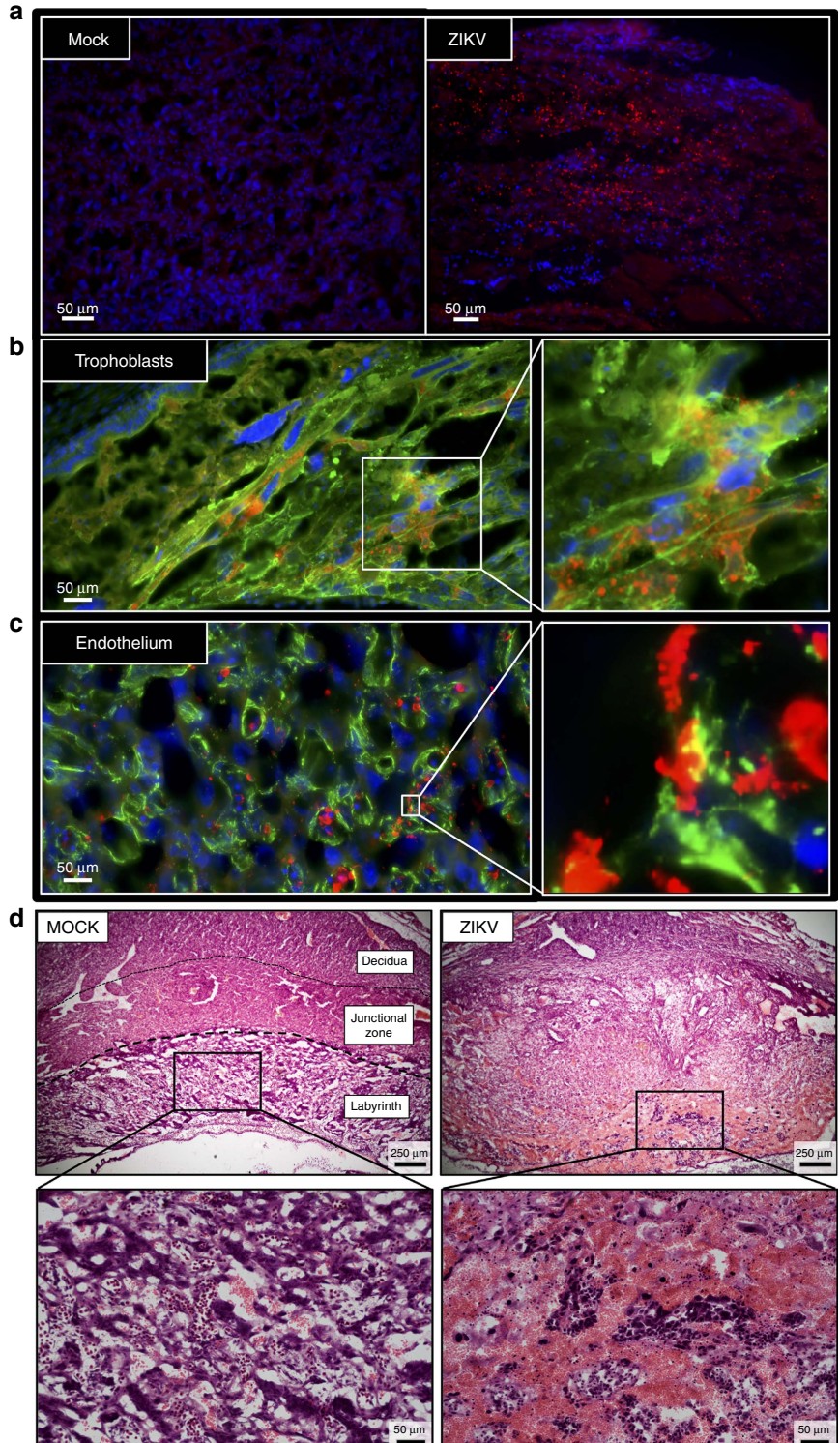

**Figure 4 | ZIKV antigen localizes in placental trophoblast and endothelial cells.** Pregnant CD1 dams underwent a mini-laparotomy in the lower abdomen for intrauterine (IU) inoculation of ZIKV (1968 Nigeria) or vehicle at embryonic day 10. Placentas were harvested 48 h post-inoculation for immunohistochemical analysis. (**a**) Fluorescent immuno-staining of ZIKV (red) with 4′6-diamidino-2-phenylindole (DAPI, blue) to label nuclei in mock-infected (left panel) and infected (right panel) placentas. Scale bar, 50 μm. (**b**) Fluorescent immuno-staining of ZIKV (red) with Cytokeratin (green), a trophoblast marker or (**c**) Vimentin (green), an endothelial cell marker, in placentas. DAPI (blue) was used to label nuclei. For **b,c**, the right panel is high magnification of white box in the left panel. Scale bar, 50 μm. (**d**) Representative H & E images of placenta were taken at × 5 magnification, with bottom panels being high magnification of black boxes in the top panel. Virus-inoculated placenta demonstrated loss of definition between placental layers, reduction in the size of the placental labyrinth, and significant maternal haemorrhage (non-nucleated red blood cells) in labyrinth layer of placenta (haematoxylin negative) mixed with fetal blood cells (haematoxylin positive; nucleated red cells). Scale bar: 250 μm in upper panels, 50 μm in lower panels. Representative images from n = 10 dams are shown.

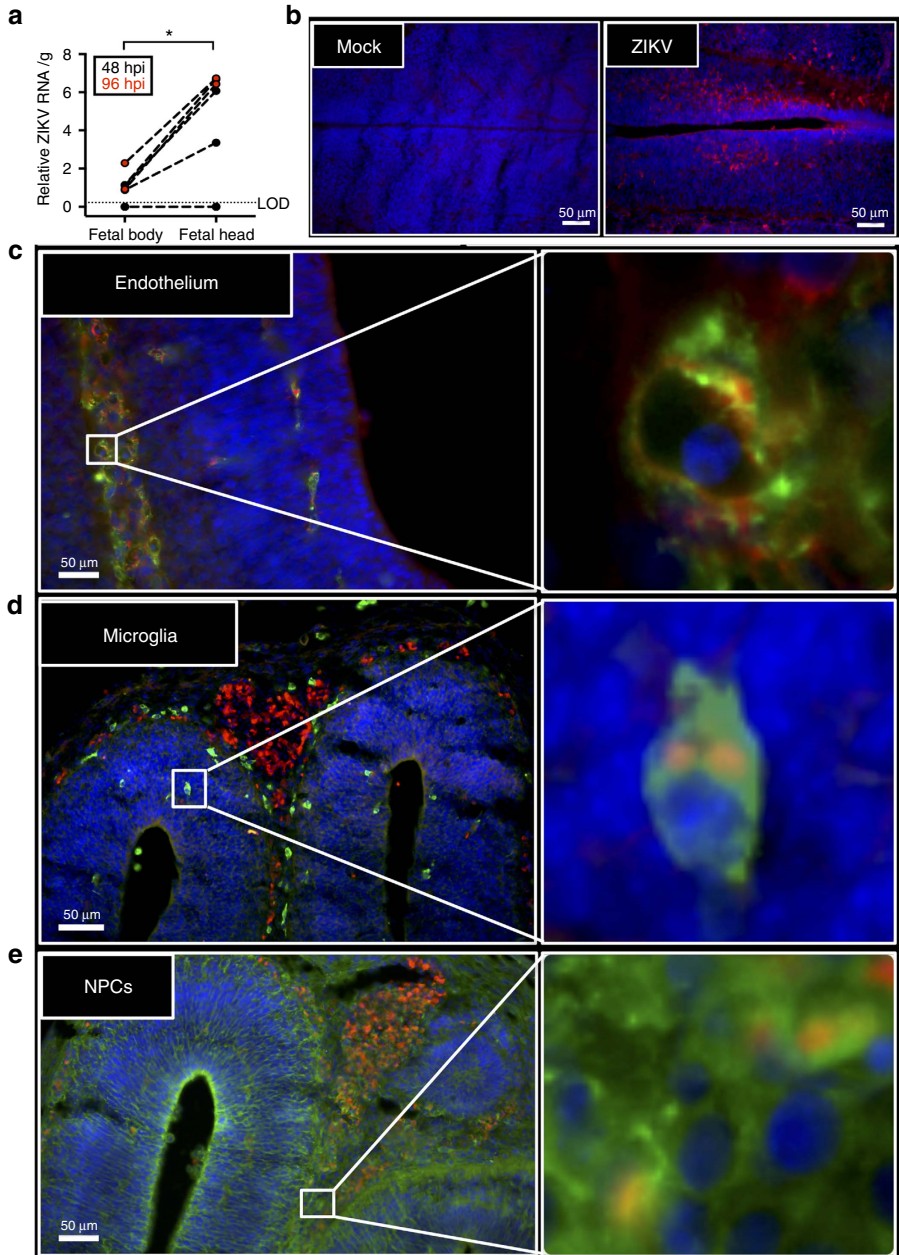

**Figure 5 | ZIKV antigen localizes in fetal brain cells.** Pregnant CD1 dams underwent a mini-laparotomy in the lower abdomen for intrauterine (IU) inoculation of ZIKV (1968 Nigeria) or vehicle at embryonic day 10. Fetal brains and bodies were harvested 48 or 96 h post-inoculation for quantification of viral RNA or immunohistochemical analysis. (**a**) ZIKV RNA was quantified in fetal bodies and heads collected 48 or 96 hpi, with the limit of detection (LOD) indicated with a dashed line. (**b**) Fluorescent immuno-staining of ZIKV (red) with 4'6-diamidino-2-phenylindole (DAPI, blue) to label nuclei in mock-infected (left panel) and infected (right panel) fetal brains. Scale bar, 50 μm. Fluorescent immuno-staining of ZIKV (red) with (**c**) CD34 (green), an endothelial cell marker, (**d**) Iba-1 (green), a microglial cell marker, or (**e**) Nestin (green), a neural stem cell marker, in fetal brains. DAPI (blue) was used to label nuclei. For **c–e**, the right panel is high-magnification of the white boxes in left panel. Scale bar, 50 μm. Representative images from $n = 10$ litters are shown. For **a**, data were analysed with a paired $t$-test, $*$ = significant difference at $P < 0.05$.

mice, and infection of these mice during early pregnancy (that is, before placentation), either systemically[18] or vaginally[15] with diverse strains of ZIKV, results in productive systemic infection and dissemination of the virus to the fetus. Studies *in vitro* highlight the role of type I IFN signalling in limiting ZIKV infection and replication in human epithelial cell lines[34,35]. To confirm activation of type I IFN signalling in our ZIKV model, we measured type I IFNs (IFNα and IFNβ) and associated IFN-stimulated genes (ISGs) in the placentas of mock and ZIKV IU-

inoculated dams at E10. At 48 hpi, concentrations of IFNα were below the limits of detection ($12.5 \, \mathrm{pg \, ml}^{-1}$), whereas concentrations of IFNβ were significantly greater in ZIKV-compared with mock-infected placentas (Fig. 6a). Furthermore, the levels of IFNβ were positively correlated with the relative ZIKV RNA load in placentas (Fig. 6b). The relative expression of the ISG messenger RNAs (mRNAs), *Isg15, Ifitm3 and Oas1b* was significantly higher in ZIKV- compared with mock-infected placentas (Fig. 6c–e). Taken together, these data illustrate that IU

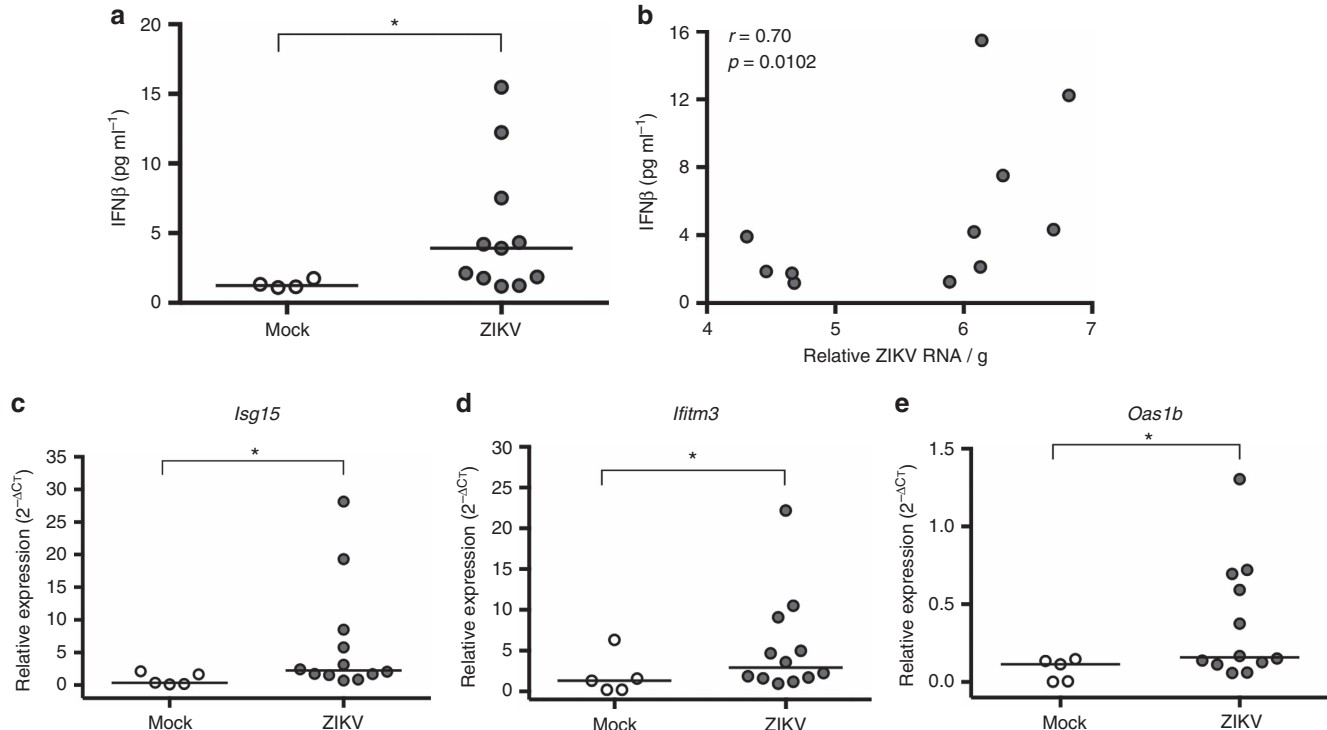

**Figure 6 | ZIKV infection induces production of type I IFNs and activation of IFN-stimulated genes (ISGs) in the placenta.** Pregnant CD1 dams underwent a mini-laparotomy in the lower abdomen for intrauterine (IU) inoculation of ZIKV (1968 Nigeria) or vehicle at embryonic day 10. Placentas were harvested 48 h post-inoculation. (**a**) The concentrations of placental IFNβ from mock- and ZIKV-infected dams were quantified from tissue homogenates by ELISA, and (**b**) correlated with relative ZIKV RNA. Placental expression of the ISG mRNAs, (**c**) *Isg15*, (**d**) *Ifitm3* and (**e**) *Oas1b* was quantified by qRT-PCR, and gene expression is presented relative to the housekeeping gene, *Gapdh*. For **a,c–e**, the median is indicated by the solid line for each group and analysed by Mann–Whitney U analyses. The data in **b** were analysed with Spearman correlation analyses, * = significant difference at $P < 0.05$.

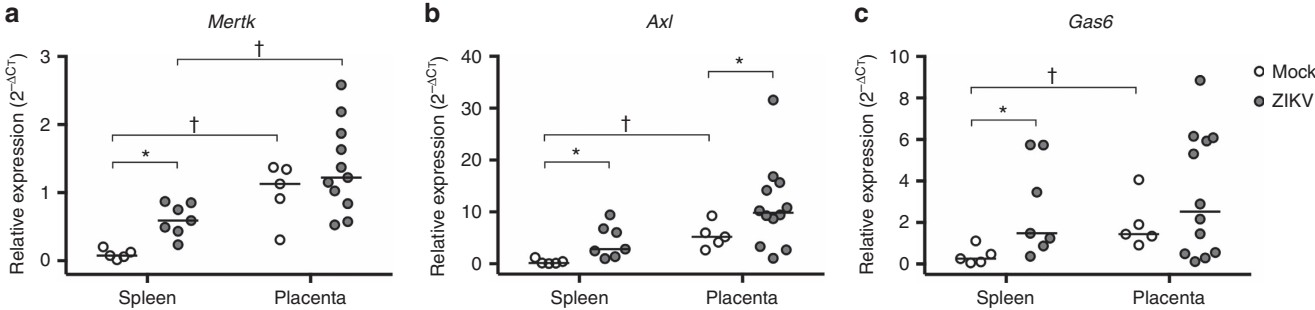

**Figure 7 | TAM receptor expression is elevated in the placenta and induced by ZIKV infection.** Pregnant CD1 dams underwent a mini-laparotomy in the lower abdomen for intrauterine (IU) inoculation of ZIKV (1968 Nigeria) or mock infection at embryonic day 10. Maternal spleens and placentas were harvested 48 h post-inoculation, and TAM receptor expression was measured by qRT-PCR. Relative (**a**) *Mertk*, (**b**) *Axl* and (**c**) *Gas6* expression was compared between the spleen and placenta of mock- and ZIKV-infected dams. For each panel, gene expression is presented relative to the housekeeping gene, *Gapdh*, with the median indicated by the solid line for each group For each panel, paired *t*-tests (spleen versus placenta) and Mann-Whitney (mock versus ZIKV) analyses, * = significant difference at $P < 0.05$ between mock and ZIKV infections. † = significant difference at $P < 0.05$ between expression levels in paired spleen and placentas within an infection group.

ZIKV infection in immunocompetent mice results in the induction of type I IFN and associated downstream targets in the placenta.

**TAM receptor tyrosine kinases in the placenta.** Although the entry receptor(s) for ZIKV has not been identified, studies *in vitro* highlight TAM receptor tyrosine kinases as one class of proteins that can mediate ZIKV entry. Within the TAM receptor family, *Axl* has been detected on multiple ZIKV cellular targets, including

subsets of placental trophoblasts[36]. To characterize the role of TAM receptors *in vivo*, we measured relative mRNA expression of the receptors *Mertk* and *Axl*, as well as the associated adaptor protein, *Gas6*, in maternal spleens and placentas from both mock- and ZIKV-infected dams (Fig. 7a–c). In mock-infected dams, relative expression of *Mertk*, *Axl* and *Gas6* was significantly greater in the placenta compared with the spleen. In the spleen, relative expression of *Mertk*, *Axl* and *Gas6* was upregulated in ZIKV-infected compared with mock-infected dams. In the placenta, the expression of *Axl* was upregulated in

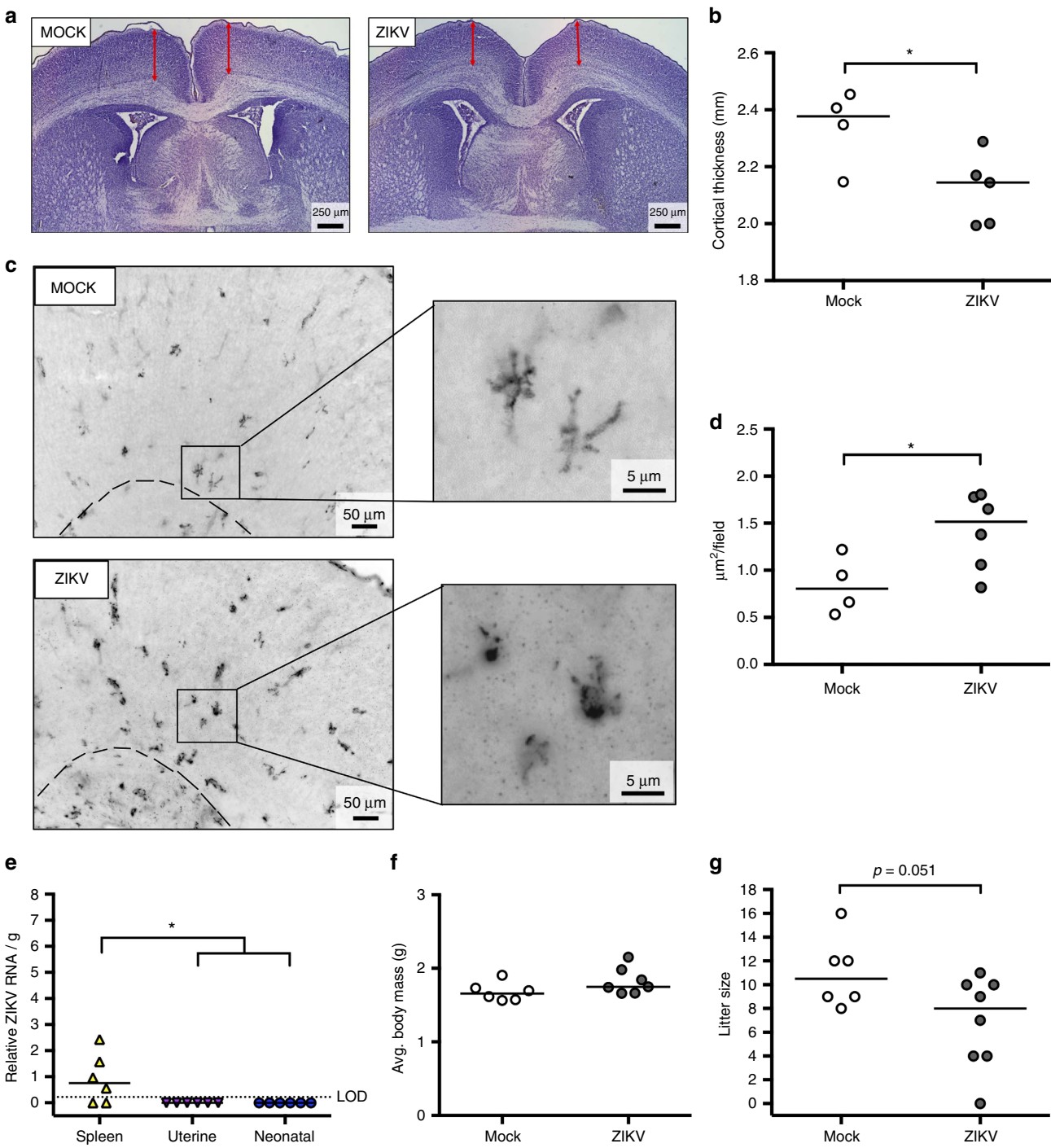

**Figure 8 | ZIKV infection results in neuroinflammation and cortical thinning in neonatal brains.** At embryonic day (E) 10, pregnant CD-1 mice underwent a mini-laparotomy in the lower abdomen for intrauterine (IU) inoculation of ZIKV (1968 Nigeria, $n = 8$ litters) or mock ($n = 6$ litters) infection. Dams were monitored until delivery. At postnatal day (PND) 0, dams and neonates were killed and pups were counted and weighed. Fetal cortical thickness was measured from Nissl-stained sections of neonatal brains. (**a**) Cortical thickness (red arrows) was measured from both brain hemispheres in neonates from both mock- and ZIKV-infected dams. Scale bar, 250 μm. (**b**) Quantitative data, with the median indicated by the solid line for each group, represent the average of these measurements from five random brain sections for each litter, in order to avoid litter effects and type II error ($n = 4$–5 litters/treatment). (**c**) Immunohistochemistry of neonatal brains was performed to detect Iba-1+ microglia as a marker of neural inflammation. Scale bar, 50 μm in left panels and 5 μm in right. (**d**) The concentration of Iba-1+ microglia per ×20 magnification field was calculated and the solid lines indicate the group means. (**e**) ZIKV RNA was quantified from maternal spleens, uterine horns and neonatal heads, with the median indicated by the solid line for each tissue and the limit of detection indicated with a dashed line. (**f**) Mean neonatal body mass was determined for each litter and (**g**) average litter size was calculated. For **b**, **d**,**f**,**g**, Mann–Whitney U tests were used and for **e**, one-way ANOVA was conducted, * = significant difference at $P < 0.05$.

ZIKV-infected dams, but *Mertk* and *Gas6* remained unchanged, compared with mock-infected dams. Taken together, these data suggest that relative TAM receptor expression is tissue-dependent, and expression can be induced with ZIKV infection.

**Postnatal neuroinflammation and cortical thinning**. To characterize the effects of *in utero* ZIKV infection on fetal development, a subset of dams infected at E10 were followed to term. At postnatal day (PND) 0, neonates were killed, weighed and tissues were collected for detection of virus and histological evaluation of the brain. Cortical thickness was quantified from Nissl-stained tissue sections of neonatal brains (Fig. 8a) as a measure of neural development[14,30], and correlate of cortical brain injury as detectable by diffusion magnetic resonance imaging[37]. Immunohistochemistry was also performed to detect and quantify Iba-1 + microglia as a marker of neural inflammation. At PND 0, compared with those from mock-infected dams, neonatal brains from ZIKV-infected dams had reduced cortical thickness (Fig. 8a,b), increased concentrations of Iba-1 + microglia (Fig. 8c,d), and evidence of increased microglia activation as shown by their amoeboid shape (Fig. 8c). Cortical thinning and neural inflammation (as depicted by the amoeboid shape of the microglia in Fig. 8c) was observed in the absence of detectable ZIKV RNA (Fig. 8e). Furthermore, there was no detectable reduction in body mass of neonates from ZIKV-infected dams (Fig. 8f), despite liter sizes being smaller in ZIKV- compared with mock-infected dams (Fig. 8g). These data illustrate that IU ZIKV inoculation into pregnant immunocompetent mice results in neuroinflammation and cortical thinning in postnatal brains.

**Discussion**
Epidemiological studies illustrate that ZIKV infection during pregnancy can cause congenital infection and fetal abnormalities[11]. However, not all pregnant women infected with ZIKV will transmit the virus to the fetus, and not all infected fetuses will exhibit overt evidence of congenital disease[12,13,38–43]. The pathogenesis of ZIKV infection at the maternal–fetal interface and the mechanisms that mediate transplacental transmission, however, are poorly understood. We demonstrate that IU inoculation with multiple strains of ZIKV into pregnant immunocompetent mice results in placental and fetal infection and adverse perinatal outcomes. Using this model, we have characterized the pathogenesis of infection with diverse ZIKV strains at the maternal–fetal interface, identified gestational age as a factor that facilitates transplacental ZIKV transmission, and investigated postnatal sequelae.

Direct inoculation of infectious agents into the reproductive tract of pregnant mice has been established as a model for studying complications associated with IU inflammation[44], ascending bacterial infections[45] and teratogenic viral infections[46] during pregnancy. In the context of our model of ZIKV infection of pregnant mice, direct inoculation was necessary for viral dissemination and replication in placentas and fetuses. In our study, systemic IP inoculation with an equivalent dose of ZIKV did not result in detectable ZIKV RNA or infectious virus placentas or fetuses. Similar to *in vitro* cell cultures and organoid systems[23,36,47–50], this model permits controlled delivery of ZIKV to reproductive tissues for the focused study of virus pathogenesis at the maternal–fetal interface. The advantage of this model over *in vitro* cultures and organoid systems is that ZIKV infection can be analysed in the context of pregnancy and an intact immune response.

The association between ZIKV infection during pregnancy and adverse perinatal outcomes has been better documented with contemporary than with historic ZIKV strains. We compared relative infectivity, transmission rates and disease outcomes following IU inoculation with both historic African (1968 Nigeria) and contemporary Asian (2010 Cambodia), and American (2015 Brazil and 2015 Puerto Rico) ZIKV isolates. Following IU inoculation, there were no significant differences in relative viral load, transmission rate or fetal outcome between different ZIKV strains. Although it is possible that differences in systemic antiviral responses to the various ZIKV strains may influence virus dissemination, upon entry into the female reproductive tract during pregnancy, the potential for maternal–fetal transmission is conserved across diverse ZIKV strains. These data are supported by recent clinical observations in Southeast Asia illustrating that ZIKV isolates from outside of the Americas are associated with the induction of human birth defects[51] and that both African and Asian lineages of ZIKV have the capacity to infect the reproductive tract of male mice[52]. Our observation that the mouse-adapted 1968 Nigerian strain caused altered perinatal outcomes to the same extent as Asian and American strains suggests that all strains of ZIKV may possess the ability to cross the placenta and lead to adverse perinatal outcomes and long-term sequelae. The passage history of the 1968 Nigeria ZIKV (IBH30656) in suckling mice may result in more efficient virus replication than the other strains (for example, 2010 Cambodia, 2015 Brazil and 2015 Puerto Rico) that we tested, suggesting that translation of these findings to other strains of ZIKV remains to be determined[53,54].

The placenta is an important physical and immunological barrier against transmission of infectious agents during pregnancy. Placentation in both humans and mice is invasive and results in the positioning of placental trophoblasts in direct contact with maternal blood. Studies *in vitro* reveal that ZIKV infects human primary cytotrophoblasts[25] and trophoblast cell lines[22,23], which is hypothesized to contribute to virus dissemination and fetal infection. We show that IU ZIKV inoculation of pregnant immunocompetent mice, after complete placentation, resulted in productive virus replication in placentas, *in vivo* localization of ZIKV antigen in placental trophoblasts and transplacental transmission of ZIKV to fetuses. Placental ZIKV infection was also associated with the induction of anti-viral inflammatory responses, including IFNβ and several ISGs, emphasizing that evaluation of type I IFN signalling is necessary to accurately study the pathogenesis of ZIKV infection at the maternal–fetal interface.

Although the entry receptor(s) for ZIKV remain unknown, many flaviviruses, including ZIKV, are thought to attach to or enter target cells by interacting with members of the TAM receptor kinase family[34,36,55–58]. The TAM receptor, *Axl*, is expressed on subsets of placental trophoblasts[12], and overexpression in primary human fibroblasts[34] and endothelial cells[58] augments ZIKV cellular entry and replication. Our data show that *Axl* expression was higher in the placentas than spleens of pregnant mice and was induced by ZIKV infection, which may suggest a mechanism for selective tissue tropism.

Congenital infections in both humans and animals are often associated with a window of susceptibility, which corresponds to a time during gestation at which the infectious agent is most likely to transmit to the fetus and have adverse effects. The specific window of susceptibility varies for different infectious agents, and is based on both pathogen and host factors. Host determinants include a complex interplay between cellular, molecular, and anatomic factors at the maternal–fetal interface, which change throughout gestation and influence the relative receptivity of placental and fetal tissues to infection. For teratogenic flaviviruses, including Japanese encephalitis virus in swine and bovine viral diarrhoea virus in cattle, this window occurs during early and mid-gestation, respectively[59]. Similarly, the highest risk of

congenital disease following ZIKV infection during pregnancy in women occurs at the end of the first trimester[12,13,40].

In immunocompetent mice, ZIKV infection and transmission following IU inoculation was dependent on gestational age, in which IU inoculation earlier during pregnancy (that is, E10) resulted in higher rates of fetal infection and resorption compared with inoculation later in gestation (that is, E14). Although E10 represents mid pregnancy in mice, the hallmarks of murine placental development at this gestational age are most similar to first trimester pregnancy in humans. E10 in mice marks the transition to the layered structure of the definitive placenta, at which time all subsets of placental trophoblast cells are present[28]. This is relatively delayed compared with placentation in humans, in which the definitive placental structure forms by 3–4 weeks post-conception (wpc)[60], and development of the maternal blood supply to the placenta is complete by the end of the first trimester ($\sim 12$ wpc)[61]. In mice and humans, the placenta continues to grow throughout gestation in order to meet the demands of the developing fetus; in CD1 mice, E14 marks the cessation of placental growth[62,63], which corresponds with the third trimester placenta in humans[64]. Although we observed reduced viral burden, transplacental transmission and fetal mortality following inoculation at E14 compared with E10, the mechanisms that underlie differences in ZIKV pathogenesis remain unknown. The extensive placental damage following ZIKV infection at E10, suggests that there are differences in either placental receptivity to infection (for example, receptor expression), placental anti-viral defenses or both. Future studies will characterize the cellular, immunologic and anatomic dynamics at the maternal–fetal interface that define the window of susceptibility for ZIKV-associated congenital infection and disease.

Congenital disease associated with ZIKV infection is characterized by a variety of fetal abnormalities, including intrauterine growth restriction (that is, fetuses that are small for gestational age), microcephaly, intracranial calcifications and enlarged ventricles[65,66]. Published data using both in vivo and in vitro systems demonstrate that ZIKV infects neural progenitor cells and triggers apoptosis that leads to cortical thinning and reduced brain size[14,22,30,67]. In concordance with these studies, our data demonstrate that IU ZIKV infection of dams at E10 resulted in ZIKV infection of the fetal brain, localization of viral antigen in fetal neural progenitor cells, increased neural inflammation and reduced cortical thickness in neonates. Viral antigen also localized in fetal endothelial and microglial cells, suggesting that multiple cell types in the fetal brain support in vivo ZIKV infection and may contribute to the observed neurologic deficits. Cortical thinning and neuroinflammation were observed in neonatal brains despite the absence of detectable viral RNA. Whether the inflammation and cortical thinning in neonatal brains was caused directly by ZIKV infection or indirectly as a consequence of placental inflammation or dysfunction remains to be determined. The observation of neuroinflammation itself, however, is evidence of neurologic damage in the offspring of infected dams, and suggests the possibility of long-term sequelae, including neurocognitive and behavioural changes[68].

The ZIKV epidemic poses a continued threat to pregnant women. In the absence of an available vaccine, diverse animal models are needed to better understand the pathogenesis of this disease. The translational potential of our mouse model of ZIKV infection is highlighted by the high viral burden detected in the placentas and corresponding fetal heads, as well as the association of type I IFN responses and ZIKV RNA at the maternal–fetal interface. Because CD1 mice have an intact immune system and genetic heterogeneity that is more similar to humans, we can better identify immunological biomarkers of disease. Translation of this infection model into different mouse strains may facilitate identification of host factors that permit higher rates of congenital infection that may function as therapeutic targets. This model uniquely strengthens the repertoire of existing in vitro and in vivo ZIKV infection models as a platform for testing early detection methods, as well as candidate vaccines and therapeutics aimed at preventing or limiting vertical ZIKV transmission or reducing the severity of congenital disease.

## Methods

**Ethics statement.** This study was carried out in accordance with the recommendations in the Guide for the Care and Use of Laboratory Animals of the National Institutes of Health. The protocols were approved by the Institutional Animal Care and Use Committee and Health Safety and the Environment at Johns Hopkins University.

**Zika virus.** The following strains of Zika virus (ZIKV) were purchased from the American Type Culture Collection (ATCC): strain IB H 30656 (Nigeria, 1968) and PRVABC59 (Puerto Rico, 2015). The FS13025 (Cambodia, 2010) strain was kindly provided by Dr Robert Tesh at UTMB, and the Pariaba (Brazil, 2015) strain was kindly provided by Dr Stephen Whitehead at the National Institute of Allergy and Infectious Diseases. All procedures for handling ZIKV were approved by the Institutional Biosafety Council. The ZIKV Nigeria 1968 virus was sequenced and confirmed to be 99.99% identical to the virus sequence in Genebank (Genebank accession number HQ234500).

**Cells and infections.** Vero cells (African green monkey kidney epithelial cells) were obtained from low-passaged seed stocks authenticated and verified to be mycoplasma-free from Dr Andrew Pekosz. Only stocks of frozen cells within two passages of the original source material were used. All cells were maintained in Dulbecco's Modified Eagle Medium (DMEM) supplemented with 10% fetal bovine serum, 1 mM sodium pyruvate, 100 U ml$^{-1}$ penicillin, 100 µg ml$^{-1}$ streptomycin and 2 mM L-glutamine. The cells were cultured in humidified air supplemented with 5% $CO_2$ at 37 °C.

Stocks of ZIKV were generated by infecting Vero cells at a multiplicity of infection (MOI) of 0.01 50% tissue culture infectious doses ($TCID_{50}$) per cell in DMEM supplemented with 2.5% fetal bovine serum, 1 mM sodium pyruvate, 100 U ml$^{-1}$ penicillin, 100 µg ml$^{-1}$ streptomycin and 2 mM L-glutamine (infection media, IM). Approximately 72 hpi, the infected cell supernatant fluids were collected, clarified by centrifugation at 900g for 10 min and stored in aliquots at $-80$ °C.

Infectious virus was quantified by $TCID_{50}$ assay. Ten-fold serial dilutions of supernatant fluids were prepared in IM and used to inoculate Vero cells in 96 well tissue culture dishes (6 wells per dilution). The cells were incubated at 37 °C for 5–6 days at which time an equal volume of 4% formaldehyde solution was added to each well and incubated for 15 min at room temperature to inactivate infectious virus. The inactivated media was removed and the cells stained with naphthol blue–black solution. Infectious virus titres were determined by the method of Reed and Muench[69].

**Mouse experiments.** Timed-pregnant adult (2–3 months of age) CD1 mice were purchased from Charles River Laboratories. Animals arrived at E9 and were housed in a specific-pathogen-free facility at Johns Hopkins University with ad libitum access to food and water. Dams were allowed to acclimate for >24 h, which is sufficient time for most basic measures of stress and immune function to return to baseline in CD-1 mice[70]. All experimental procedures were performed at the same time of the day.

At either E10 or E14, mice were anesthetized continuously with isoflurane and underwent a mini-laparotomy in the lower abdomen for ZIKV injections. Animals were randomly assigned to receive either $10^6$ $TCID_{50}$ units of ZIKV suspended in 100 µL DMEM or 100 µL DMEM alone. The dose of ZIKV was based on pilot studies in which IU infection of dams with doses <$10^6$ $TCID_{50}$ per ml did not result in placental or fetal infection. For animals designated for IP injection, the full inoculum was delivered into the abdominal cavity. For animals designated for IU injection, the inoculum was divided equally into four injections delivered into the uterine myometrium, opposite the placenta and between the gestational sacs of the first five fetuses closest to the cervix of one uterine horn. The contralateral uterine horn was not manipulated. Routine closure was performed after injections and dams were returned to individual cages for recovery. Investigators were not blinded to the treatment group allocation.

Mice were randomly selected to be killed by $CO_2$ exposure followed by cardiac exsanguination at either 48 (E12 or E16) or 96 (E14 or E18) hours post-infection or after delivery postnatal (PND) 0, depending on experimental group. At the time of euthanasia, the total number of viable and aborted/resorbed fetuses were quantified for each dam. Fetal viability was determined as the percentage of fetuses within the inoculated uterine horn for ZIKV- and mock-infected dams that were viable. Small or discoloured fetuses, or the absence of a fetus at an implantation site were counted as aborted or resorbed fetuses, respectively. Maternal spleen, serum, uterine horns,

placentas and fetuses were collected from each dam for determination of viral burden by qRT-PCR, $TCID_{50}$ assay and immunohistochemistry. All animal studies were conducted under animal BSL2 containment.

**Tissue harvesting.** Tissues were weighed and homogenized in a 1:10 weight per volume of PBS using Lysing Matrix D tubes (MP Biomedicals) in a MP Fast-prep 24 5G instrument. Tissue homogenates were stored at $-80\,°C$. Total RNA was extracted from tissue samples using the RNeasy Lipid Tissue Mini Kit (Qiagen) or from serum and cell culture supernatant using QIAamp Viral RNA Mini Kit (Qiagen), according to the manufacturer's instructions.

**Quantitative RT-PCR.** Total RNA was extracted from tissue samples using the RNeasy Lipid Tissue Mini Kit (Qiagen) or from serum and cell culture supernatant using QIAamp Viral RNA Mini Kit (Qiagen).

ZIKV RNA levels were determined by one-step quantitative reverse transcriptase-PCR (qRT-PCR) reaction using the QuantiTect Probe RT-PCR Kit (Qiagen) according to the manufacturer's protocol. The real-time PCR primers and probe for ZIKV RNA detection were the ZIKV 1162c set described previously[71], which recognize diverse ZIKV strains but with limited sensitivity[72]. Primer sequences and concentrations were as follows: Fwd (100 µM), 5′-CCGCTGCC CAACACAAG-3′; Rev (100 µM), 5′-CCACTAACGTTC TTTTGCAGACAT-3′; Probe (25 µM), 5′-/56-FAM/AGCCTACCT/ZEN/TGACAAGCAATCAG ACACTCAA/3IABkFQ/-3′ (Integrated DNA Technologies). ZIKV RNA copies were determined relative to a standard curve produced using serial 10-fold dilutions of ZIKV (1968 Nigerian) RNA isolated from ZIKV stocks with a known infectious virus titre. Relative levels of viral RNA in tissue were similar whether values were calculated based on the $\log_{10}$ dilution curve for ZIKV RNA or relative to the host housekeeping gene, *Gapdh* (Pearson correlation, $R^2 = 0.913$, $P < 0.0001$).

For TAM receptors and ISGs, RNA was quantified using a nanodrop spectrophotometer, and complementary DNA (cDNA) was prepared with the High Capacity cDNA reverse transcription kit (Life Technologies) using 2.5 µg of input RNA. For the TAM receptors (*Mertk, Axl* and *Gas6*), quantitative real-time PCR (qRT-PCR) was performed using 2.5 µl of cDNA, TaqMan gene expression arrays (*Mertk,* Mm00434920_m1; *Axl,* Mm00437221_m1; and *Gas6,* Mm00490378_m1) and $\times 2$ Universal PCR Mastermix (Applied Biosystems), according to manufacturer's instructions. For the ISGs (*Isg15, Ifitm3* and *Oas1b*), primers were purchased from Integrated DNA Technologies, with sequences as follows: *Isg15:* Fwd, 5′-GGTGTCCGTGACTAACTCCAT-3′; Rev, 5′-TGGAAAGGGTAAGAC CGTCCT-3′; *Ifitm3:* Fwd, 5′-CCCCCAAACTACGAAAGAATCA-3′; Rev, 5′-AC CATCTTCCGATCCCTAGAC-3′; *Oas1b:* Fwd, 5′-GGGCCTCTAAAGGGGTCA AG-3′; Rev, 5′-TCAAACTTCACTCCACAACGTC-3′. Quantitative PCR was performed using the SsoFast EvaGreen Supermix (Biorad) according to manufacturer's instructions. All reactions were run on the StepOnePlus Applied Biosystems Real-time PCR machine under the following conditions: 50 °C for 2 min, 95 °C for 10 min, 95 °C for 15 s and 60 °C for 1 min for 50 cycles. Transcript levels were determined by normalizing the target gene threshold cycle (CT) value to the CT value of the endogenous housekeeping gene Gapdh (ΔCT).

**ELISA.** Interferon-α (IFN-α) and IFN-β levels were measured by enzyme-linked immunosorbent assay (ELISA) (Verikine ELISA kits; PBL Interferon Source, Piscataway, NJ) according to the manufacturer's instructions. Placentas (10%, wt per vol) were tested in duplicate. Sensitivity was 12.5–400 pg ml$^{-1}$ IFN-α and 0.94–60 pg ml$^{-1}$ for IFN-β.

**Immunohistochemistry and immunofluorescence imaging.** Tissues were fixed in 4% paraformaldehyde for 24–48 h at 4 °C. Tissues were washed with PBS extensively and immersed in 30% sucrose until saturation, followed by cryosection at 20 µm thickness. Routine hematoxylin and eosin (H&E) staining was performed to evaluate the morphological change of the placentas. Antigens were retrieved by boiling in PBS buffer for 20 min. After antigen retrieval, tissues were blocked in 10% goat serum and permeabilized with 0.5% Triton-X-100. Tissues were incubated with primary antibodies overnight at 4 °C. The following primary antibodies were used: mouse (1:200, MAB10216, clone D1-4G2-4-15, EMD Millipore, Billerica, MA) or rabbit (1:200, Ab00230-23.0, clone D1-4G2-4-15, Absolute Antibody, Oxford, UK) antibodies to flaviviruses to identify ZIKV; rabbit anti-cytokeratin (1:200, Z0622, DAKO, Carpinteria, CA) to identify trophoblasts, rabbit anti-vimentin (1:200, ab92547, clone EPR3776, Abcam, Cambridge, MA) to identify fetal capillaries, rabbit anti-α-actin (1:200, ab5694, Abcam) to identify uterus tissue, mouse anti-nestin (1:100, sc-58813, clone 2Q178, Santa Cruz, Dallas, Texas) to identify neural stem cells, rabbit anti-Iba-1 (1:100, 019-1974, WAKO, Richmond, VA) to identify microglial cells, and rabbit anti-CD34 (1:100, ab81289, clone EP373Y, Abcam) to identify endothelial cells. The next day tissues were washed with PBS, followed by secondary antibody incubation for 3 h at room temperature. Secondary donkey antibodies used were: anti-rabbit IgG (1:500, A21206, Life Technologies, Frederick, MD) or anti-mouse IgG (1:500, ab150105, Abcam) Alexa Fluor 488; and anti-mouse IgG (1:500, A10037, Invitrogen, Rockford, IL) or anti-rabbit IgG (1:500, A10042, Invitrogen) Alexa Fluor 568. DAPI (4′,6-diamidino-2-phenylindole, 10236276001, Roche, Indianapolis, IN) was used

to counter stain nuclei at a concentration of 1:5,000 Slides were mounted with Fluoro-mount G (SouthernBiotech, Birmingham, AL) and viewed using a Zeiss Axioplan 2 microscope (Jena, Germany) with a $\times 20$ objective. Images were taken using a Zeiss AxioCam MRM. Iba-1+ microglia were quantified using ImageJ (NIH) software.

**Neonatal cortical measurements.** Neonatal mice were killed at PND 0 and heads were fixed and sectioned as described above. Routine Nissl staining was performed, and images were taken under $\times 5$ magnification using a Canon EOS Rebel (Tokyo, Japan). Coronal cortical thickness was measured from five random sections at the striatum level of each neonatal brain, as previously described[37]. Cortical thickness was measured from both brain hemispheres in each section using ImageJ software, and the average of 10 measurements per specimen presented. For quantification of cortical microglial concentrations, the area of Iba-1+ staining in the cortex was quantified from five $\times 20$ fields within the stratum level of each brain using ImageJ software. Quantification shown represents the average measurement from a single randomly chosen pup for each dam.

**Statistical analysis.** All data were analysed with GraphPad Prism software. Fetal viability data were assessed with a Chi-square test. Infectious virus titres and viral RNA data were analysed with one- or two-way ANOVAs using Kruskal-Wallis or Bonferroni *post-hoc* corrections for multiple comparisons. Correlations were analysed using a Pearson or Spearman correlation test. Interferon-β, ISG, and TAM receptor data were analysed with either a paired *t* test or Mann-Whitney test, depending on distribution. Mean cortical thickness, neonatal body mass and litter size were analysed using a Mann–Whitney test. Mean differences were considered statistically significant if $P < 0.05$.

**Data availability.** All relevant data are available from the authors on request.

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

## Acknowledgements

We thank members of the Klein and Pekosz labs for detailed discussions about these data, Anne Jedlicka for assistance with qRT-PCR and Jennifer Fairman and Alan Burd for graphic assistance. The work was supported by the Center for Emerging and Infectious Diseases (CEVID) at Johns Hopkins University, Medtronics award from the Society for Women's Health Research (S.L.K.), Integrated Research Center for Fetal Medicine (I.B.) and Sheikh Abdullah Bugshan Fund (I.B.), the NIH T32 OD011089 Training Veterinarians for Careers in Biomedical Research (M.S.V. and V.K.B.) and the Merial Veterinary Scholars Program (N.P.C.).

## Author contributions

M.S.V., J.L., S.L.K. and I.B. conceived of the idea; M.S.V., J.L., V.K.B., D.E.G., S.L.K., and I.B. designed the experiments, M.S.V., J.L., Y.S., V.K.B., N.P.C., M.M., A.P. and I.B. conducted the experiments; M.S.V., J.L., A.P. and S.L.K. analysed and graphed data; M.S.V., A.P., S.L.K. and I.B. wrote the manuscript; M.S.V., J.L., D.E.G., A.P., S.L.K. and I.B. edited the manuscript; all authors reviewed and approved the final draft of the manuscript.

## Additional information

**Competing financial interests:** The authors declare no competing financial interests.

