## [Peer Review File · Nature Communications]

Reviewers' comments:

Reviewer #1 (Remarks to the Author):

This (revised) paper from Vermillion et al., describes the development of an intrauterine model of ZIKV infection during early placentation. They have now tested a number of contemporary Zikv strains in this model, and identified a number of signaling molecules and targets that may be relevant to understanding ZIKV pathogenesis at the maternal fetal interface.

I have no remaining concerns.

Reviewer #2 (Remarks to the Author):

This is a revised manuscript describing a mouse model of ZIKV infection of pregnant outbred CD-1 mice. The authors have responded to the previous reviewers comments. However, several questions still remain. Overall, it is not clear how this mouse model of ZIKV infection provides a significant advancement into our understanding of the mechanisms of transplacental transmission of ZIKV.

Comments:

1) The authors should show ZIKV RNA detection in the serum, rather than leaving it as data not shown.

2) What accounts for the differences in virus replication following infection of either E10 or E14 infected pregnant mothers? While phenotypically the authors show reduced virus replication, the reason for this finding is not entirely clear. Why are the tissues more resistant to ZIKV replication? Is this linked to STAT2 expression within the uterus?

3) The authors over-stated their response to reviewer 2 in that they are the first group to show ZIKV localization with cytokeratin+ trophoblasts. See Miner et al paper. Furthermore, the authors have not satisfactorily addressed whether trophoblasts are infected in pregnant mothers. There is no histological evidence to support their findings in the mouse model. The authors argue that trophoblast cell lines can be infected by ZIKV. Human A549 cells, a lung carcinoma cell line, can also be infected by ZIKV, but that does not mean that lungs are infected by ZIKV. There are many types of trophoblasts (as mentioned in the previous comments) and mouse and human placentas are vastly different. It is not clear how this mouse model would provide insight into human ZIKV transplacental transmission. This continues to be a major concern with regards to the findings presented in this manuscript.

4) The authors did not address the previous question about whether the mice die because of direct ZIKV replication in the brain or due to direct ZIKV replication and damage of the mouse placenta.

5) Is ZIKV replication detected elsewhere in the fetus, besides the head? In humans, ZIKV is only found in the brain. The authors should evaluate ZIKV replication in other tissues within the embryos.

6) The use of Iba-1 as a marker of neuroinflammation is a good start, however, several questions still remain. Did the authors measure cytokine production? Why is no viral antigen present in cells that stain for Iba-1? What is the cause of the neuroinflammation if it is virus-independent?

7) The measurement of cortical thickness in Figure 8b requires a larger sample size. There are only 4-5 measurements. As it stands, the current data are not sufficient to warrant an appropriate conclusion, especially with the difference in measurements between mock and ZIKV being about 10%.

Reviewer #3 (Remarks to the Author):

The manuscript presented by Vermillion et al. describes a model of intra-uterine (IU) ZIKV infection in immunocompetent CD1 mice, to study trans-placental transmission of the virus and adverse perinatal outcomes. The authors use four ZIKV strains from different origins (Nigeria 1968, Cambodia 2010, Brazil 2015, Puerto Rico 2015), analyze both intra-peritoneal (IP) and IU routes of infection and gestational age (time of complete placentation embryonic day 10 [E10] and time of cessation of placental expansion and late gestation E14). Vermillion et al. observe that when the virus is inoculated at E10, ZIKV reduces fetal viability, and there is an increase of viral infection in the placenta, uterine horns and in the fetal head, opposite to what happens at E14. They also observe placental inflammation, reduced neonatal brain cortical thickness, and production of IFN and IFN-stimulated genes after ZIKV infection at E10. Additionally, ZIKV is detected in several maternal and fetal cells by immunohistochemistry (trophoblast and endothelial cells in the placenta, and endothelial, microglial and neural progenitor cells in the fetal brain). In this revised version, the authors additionally analyze the level of putative ZIKV entry receptors, and observe that their levels increase upon ZIKV infection in the placenta.

The manuscript is well presented, correctly written and the authors have addressed most of the reviewers' concerns, so the quality of the paper has increased. Nevertheless, there is an additional major point that should be addressed by the authors: As previously noted by another reviewer, the African strain used in the study, IB H 30656 (Nigeria 1968), has been extensively passaged in suckling mice (1), which might have added many mutations from the original African strain, therefore affecting the potential pathology of this strain. It has actually recently been shown a difference in neural infectivity in vitro between two low passage African and Asian strains (2), results that differ i.e. from the extensively used African strain MR766, also extensively passaged in suckling mice. Therefore, the use of this

African strain (IB H 30656) for a comparison between African and Asian&American ZIKV strains, could lead to a misleading result. An African isolate with lower number of passages would be more correct for a comparison. Otherwise, authors should recognize the caveat regarding their comparison of the effects between African, Asian and American strains of ZIKV.

1. Haddow AD, Schuh AJ, Yasuda CY, Kasper MR, Heang V, Huy R, Guzman H, Tesh RB, and Weaver SC. Genetic characterization of Zika virus strains: geographic expansion of the Asian lineage. *PLoS Negl Trop Dis*. 2012;6(2):e1477.
2. Simonin Y, Loustalot F, Desmetz C, Foulongne V, Constant O, Fournier-Wirth C, Leon F, Moles JP, Goubaud A, Lemaitre JM, et al. Zika Virus Strains Potentially Display Different Infectious Profiles in Human Neural Cells. *EBioMedicine*. 2016.

Response to reviewers

We thank the reviewers for their thorough review of our manuscript. We have made several modifications to the manuscript, including the addition of ZIKV RNA quantification in maternal serum and fetal bodies. We also evaluated placental damage as a possible mechanism for reduced fetal viability and neonatal neuroinflammation. We strengthened our discussion about the significance of our model for understanding transplacental transmission of ZIKV in immunocompetent, as opposed to immunocompromised, mice. Our specific responses to each of the reviewer's comments are detailed below.

Reviewers' comments:

Reviewer #1 (Remarks to the Author):

This (revised) paper from Vermillion et al., describes the development of an intrauterine model of ZIKV infection during early placentation. They have now tested a number of contemporary Zikv strains in this model, and identified a number of signaling molecules and targets that may be relevant to understanding ZIKV pathogenesis at the maternal fetal interface. I have no remaining concerns.

We thank the reviewer for their encouraging remarks.

Reviewer #2 (Remarks to the Author):

This is a revised manuscript describing a mouse model of ZIKV infection of pregnant outbred CD-1 mice. The authors have responded to the previous reviewers comments. However, several questions still remain. Overall, it is not clear how this mouse model of ZIKV infection provides a significant advancement into our understanding of the mechanisms of transplacental transmission of ZIKV.

We appreciate that the review has acknowledged the rigor with which we have addressed the previous concerns. In answer to the query as to how this model provides a significant advancement, we make three points in the

Introduction on p. 4 and Discussion on p. 15. First, this is the first model that enables significant amounts of ZIKV RNA and infectious virus to be detected in wild type mice with an intact immune system. Prior to this study, publications using other modes of infection in wild-type pregnant mice have either detected low to non-detectable levels of viral RNA and infectious virus or used a viral inoculum that was six logs higher than what we used to get detectable virus in wild-type animals. Second, we can dissect type I IFN activity, TAM receptor signaling, and their interaction in a way that is not possible in animals that are devoid of type I IFN signaling. Lastly, previous studies of ZIKV infection of pregnant dams have relied predominantly on infecting dams prior to complete placentation, which means that they have not adequately addressed whether in their model, virus is capable of crossing a complete placental barrier to infect fetuses. In our model, infection at E10 allows us to state that virus is capable of crossing a complete placental barrier to infect developing fetuses.

Comments:

1) The authors should show ZIKV RNA detection in the serum, rather than leaving it as data not shown.

We thank the reviewer for this recommendation, and we have added ZIKV RNA quantification in maternal serum to Figure 1 (see new data in **Fig. 1c**).

2) What accounts for the differences in virus replication following infection of either E10 or E14 infected pregnant mothers? While phenotypically the authors show reduced virus replication, the reason for this finding is not entirely clear. Why are the tissues more resistant to ZIKV replication? Is this linked to STAT2 expression within the uterus?

Other than the observation that virus replication is reduced when dams are infected at E14 as compared with E10, we do not currently have a mechanism accounting for this difference. We state, in the Discussion on p. 14, that E14 in mice marks the cessation of placental growth and corresponds developmentally with the third trimester in humans, a time when susceptibility to ZIKV is reduced. We have, however, added data showing placental damage combined with viral burden at E10 may

contribute to the mechanism of fetal demise (see new data in **Fig. 4d**). Future studies will address in greater detail the kinetics of antiviral immune responses, viral burden, and placental dysfunction. We have no evidence that STAT2 expression in the uterus is linked to ZIKV pathogenesis, and can only reiterate that ZIKV does not remain in the uterine horns after inoculation (see Results on p. 7-8).

3) The authors over-stated their response to reviewer 2 in that they are the first group to show ZIKV localization with cytokeratin+ trophoblasts. See Miner et al paper. Furthermore, the authors have not satisfactorily addressed whether trophoblasts are infected in pregnant mothers. There is no histological evidence to support their findings in the mouse model. The authors argue that trophoblast cell lines can be infected by ZIKV. Human A549 cells, a lung carcinoma cell line, can also be infected by ZIKV, but that does not mean that lungs are infected by ZIKV. There are many types of trophoblasts (as mentioned in the previous comments) and mouse and human placentas are vastly different. It is not clear how this mouse model would provide insight into human ZIKV transplacental transmission. This continues to be a major concern with regards to the findings presented in this manuscript.

In light of recent studies that report concordant results, we have omitted the statement referenced by the reviewer in regard to ZIKV localization with Cytokeratin+ trophoblasts. We also recognize that because similar studies in humans have been very limited, ZIKV antigen has not yet been detected in trophoblast cells from placentas from infected women, and data suggesting that ZIKV infects trophoblast cells has been limited to *ex vivo* infection of placental explants. These studies, however, have been replicated by several groups using both primary placental cell cultures as well as immortalized cell lines, and the data suggest that certain subsets of human trophoblasts are susceptible to infection.

4) The authors did not address the previous question about whether the mice die because of direct ZIKV replication in the brain or due to direct ZIVK replication and damage of the mouse placenta.

In response to this query, we now include additional representative H&E

images of placentas from mock and ZIKV-infected dams at E10, showing evidence of significant placental pathology that is likely to affect the observed fetal demise (please see new images in **Fig. 4d** and corresponding text in the Results on p. 8). While it is possible that ZIKV replication in the fetal brain may also contribute to fetal death, it is difficult to tease apart the relative contributions of placental vs. fetal infection using an *in vivo* system, as infection of each of these tissues in isolation is not possible. We discuss this issue in the Discussion on p. 14.

5) Is ZIKV replication detected elsewhere in the fetus, besides the head? In humans, ZIKV is only found in the brain. The authors should evaluate ZIKV replication in other tissues within the embryos.

We thank the reviewer for this suggestion. Recent human reports indicate that ZIKV can replicate outside the brain (Melo et. al., *JAMA Neurol.* 2016). We now provide data showing that following IU inoculation at E10, viral RNA is detected at significantly lower levels in fetal bodies than heads, which forms the rationale for why we only dissected the cell types in the fetal brain where ZIKV localizes. These data are now provided in **Fig. 5a** as well in the corresponding text in the Results on p.8.

6) The use of Iba-1 as a marker of neuroinflammation is a good start, however, several questions still remain. Did the authors measure cytokine production? Why is no viral antigen present in cells that stain for Iba-1? What is the cause of the neuroinflammation if it is virus-independent?

Characterization of the inflammatory response in the fetal brain will be a focus of our future studies. It is possible that the observed neuroinflammation was a direct consequence of viral infection of the fetal brain, but virus was cleared prior to birth. Alternatively, it is possible that placental infection and dysfunction (see **Fig. 4d**) induced a dysregulated inflammatory response in the fetus in the absence of direct fetal infection. Future studies will be aimed at characterizing the kinetics of viral replication and consequential inflammatory response, including cytokines, in both the placenta and fetal brains over time during pregnancy.

7) The measurement of cortical thickness in Figure 8b requires a larger

sample size. there are only 4-5 measurements. As it stands, the current data are not sufficient to warrant an appropriate conclusion, especially with the difference in measurements between mock and ZIKV being about 10%.

In response to this critique, we would like to clarify that each of the measurements for cortical thickness represents a separate litter, which is done to control for litter effects and Type II error. In the legend of Figure 8 (p. 20), we now clarify that each symbol represents a litter. We cannot comment on the clinical relevance of a 10% reduction in cortical thickness, as this would require additional neurocognitive and behavioral analyses that are beyond the scope of this manuscript. Regardless, our quantitative analyses reveal significant differences between our treatment groups, which are consistent with previous *in vitro* and *in vivo* studies, and which we believe to be an important observation to report.

Reviewer #3 (Remarks to the Author):

The manuscript presented by Vermillion et al. describes a model of intra-uterine (IU) ZIKV infection in immunocompetent CD1 mice, to study trans-placental transmission of the virus and adverse perinatal outcomes. The authors use four ZIKV strains from different origins (Nigeria 1968, Cambodia 2010, Brazil 2015, Puerto Rico 2015), analyze both intra-peritoneal (IP) and IU routes of infection and gestational age (time of complete placentation embryonic day 10 [E10] and time of cessation of placental expansion and late gestation E14). Vermillion et al. observe that when the virus is inoculated at E10, ZIKV reduces fetal viability, and there is an increase of viral infection in the placenta, uterine horns and in the fetal head, opposite to what happens at E14. They also observe placental inflammation, reduced neonatal brain cortical thickness, and production of IFN and IFN-stimulated genes after ZIKV infection at E10. Additionally, ZIKV is detected in several maternal and fetal cells by immunohistochemistry (trophoblast and endothelial cells in the placenta, and endothelial, microglial and neural progenitor cells in the fetal brain). In this revised version, the authors additionally analyze the level of putative ZIKV entry receptors, and observe that their levels increase upon ZIKV infection in the placenta.

The manuscript is well presented, correctly written and the authors have addressed most of the reviewers' concerns, so the quality of the paper has increased. Nevertheless, there is an additional major point that should be addressed by the authors: As previously noted by another reviewer, the African strain used in the study, IB H 30656 (Nigeria 1968), has been extensively passaged in suckling mice (1), which might have added many mutations from the original African strain, therefore affecting the potential pathology of this strain. It has actually recently been shown a difference in neural infectivity in vitro between two low passage African and Asian strains (2), results that differ i.e. from the extensively used African strain MR766, also extensively passaged in sucking mice. Therefore, the use of this African strain (IB H 30656) for a comparison between African and Asian&American ZIKV strains, could lead to a misleading result. An African isolate with lower number of passages would be more correct for a comparison. Otherwise, authors should recognize the caveat regarding their comparison of the effects between African, Asian and American strains of ZIKV.

1. Haddow AD, Schuh AJ, Yasuda CY, Kasper MR, Heang V, Huy R, Guzman H, Tesh RB, and Weaver SC. Genetic characterization of Zika virus strains: geographic expansion of the Asian lineage. PLoS Negl Trop Dis. 2012;6(2):e1477.

2. Simonin Y, Loustalot F, Desmetz C, Foulongne V, Constant O, Fournier-Wirth C, Leon F, Moles JP, Goubaud A, Lemaitre JM, et al. Zika Virus Strains Potentially Display Different Infectious Profiles in Human Neural Cells. EBioMedicine. 2016.

We thank the reviewer for this reminder, and have addressed these limitations in the Discussion on p. 12, citing the suggested references. Since the review of our manuscript, another paper was published illustrating that African lineage ZIKV can productively infect the reproductive tract of male mice, further illustrating that ZIKV of diverse lineages possess the capability of infecting reproductive tissue.

REVIEWERS' COMMENTS:

Reviewer #2 (Remarks to the Author):

The authors have addressed all of the previous concerns. There are no remaining concerns with this work.

Reviewer #3 (Remarks to the Author):

In this revised version of the manuscript, Vermillion et al. have addressed most of the concerns issued by the reviewers. The manuscript is clearly written and nicely presented. Regarding previous concerns regarding the use of an extensively passaged African strain (Nigeria 1968), additional contemporary ZIKV strains have been now tested in this mouse model.

There is one additional minor comment:

- Authors need to add a description in the Legends for Figure 4d.